# Anterior and Posterior Nutcracker Syndrome Combined with May–Thurner Syndrome: First Report of This Unique Case

**DOI:** 10.3390/diagnostics13081433

**Published:** 2023-04-16

**Authors:** Francesco Tiralongo, Federica Galioto, Giulio Distefano, Stefano Palmucci, Antonio Basile, Salvatore Di Rosa

**Affiliations:** 1Radiology Unit 1, Department of Medical Surgical Sciences and Advanced Technologies “GF Ingrassia”, University Hospital Policlinico “G. Rodolico-San Marco”, University of Catania, 95123 Catania, Italy; 2U.O.C. Diagnostica per Immagini P.O. “S. Marta e S. Verera” Acireale, ASP Catania, 95123 Catania, Italy; 3Institute of Internal Medicine, Medicine Department, Vittoria Hospital, ASP Ragusa, 97100 Ragusa, Italy; 4U.O.S.D. Radiologia, P.O. “G. Di Maria” Avola, ASP Siracusa, 96100 Siracusa, Italy

**Keywords:** renal nutcracker syndrome, May–Thurner syndrome, computed tomography, cardiovascular abnormalities, diagnostic imaging

## Abstract

Background: Anterior nutcracker syndrome is defined as the compression of the left renal vein (LRV) between the superior mesenteric artery (SMA) and the aorta, whereas posterior nutcracker syndrome refers to the compression of the retroaortic LRV between the aorta and the vertebral column—the presence of the circumaortic left renal vein may predispose to “combined nutcracker syndrome”. May–Thurner syndrome consists of obstruction of the left common iliac vein caused by the crossing right common iliac artery. We report a unique case of combined nutcracker syndrome associated with May–Thurner syndrome. Case presentation: A 39-year-old Caucasian female came to our radiology unit for triple-negative breast cancer computed tomography (CT) staging. She complained of pain in hermid-back and low-back regions and intermittent abdominal pain in the left flank region. Multidetector computed tomography (MDCT) incidentally revealed a circumaortic left renal vein draining to the inferior vena cava, with bulbous dilatation of both the antero-superior and posterior-inferior branches, which was associated with pathological serpiginous dilation of the left ovarian vein with varicose pelvic veins. Axial CT imaging of the pelvis also showed compression of the left common iliac vein by the overlying right common iliac artery consistent with May–Thurner syndrome without signs of venous thrombosis. Conclusion: Contrast-enhanced CT is the best imaging modality for suspected vascular compression syndromes. CT findings showed a combination of anterior and posterior nutcracker syndrome in the left circumaortic renal vein, associated with May–Thurner syndrome, which has not previously been described in the literature.

## 1. Background

The contiguous relationships of vascular structures with other vessels or other anatomical structures can cause extrinsic compression syndromes; as reported by Granata et al., “any vessel can be compressed” [1], but in some cases, the vessels involved, and the corresponding imaging picture, may show a characteristic pattern. In these cases, speaking of “compression syndromes” is appropriate.

These include anterior and posterior nutcracker syndrome and May–Thurner syndrome [2].

May–Thurner syndrome, also known as iliac vein compression syndrome and Cockett syndrome [3], consists of obstruction of the left common iliac vein (CIV) caused by the crossing right common iliac artery (CIA). Anterior nutcracker syndrome is the compression of the left renal vein (LRV) between the superior mesenteric artery (SMA) and the aorta. In contrast, posterior nutcracker syndrome refers to the compression of the retroaortic LRV between the aorta and the vertebral column [4]. The presence of the “renal collar”, also known as “circumaortic venous ring”—an anatomic variant characterized by the presence of two LRVs passing anteriorly and posteriorly to the aorta, respectively [2,5], may predispose to “combined nutcracker syndrome”.

The presence of a circumaortic venous ring with the simultaneous compression of the anterior LRV (between the abdominal aorta and the SMA) and the posterior LRV (between the abdominal aorta and spine) has previously been reported only four times in the literature.

Our report describes a unique case, not previously reported in the literature, of a combination of anterior and posterior nutcracker syndrome and May–Thurner syndrome.

## 2. Case Presentation

We report a case of a 39-year-old Caucasian female who came to our radiology unit for a computed tomography (CT) staging. Her medical history was remarkable for triple-negative breast cancer, diagnosed by Tru-Cut biopsy, and she had not undergone medical therapy. Laboratory tests were unrevealing.

At the clinical evaluation, the patient complained of pain in her mid-back and low-back regions and intermittent abdominal pain in the left flank region. She weighed 68 kg and had a body mass index of 20 kg/m^2^.

The patient underwent multi-detector computed tomography (MDCT, Optima 64 slice, GE Healthcare) of the brain, thorax, and abdomen.

The CT scan showed a solid nodule in the right upper outer quadrant of the breast, four hepatic metastases (confirmed later with hepatic MRI), free fluid in Douglas’s pouch, and mixed lytic and sclerotic bone metastases in the thoracic and lumbar spine (confirmed later with skeletal scintigraphy and MRI).

MDCT incidentally revealed a circumaortic left renal vein draining to the inferior vena cava, with bulbous dilatation of both the antero-superior and posterior-inferior branches. The superior anterior branch was compressed between the SMA and the aorta (Figure 1); the inferior-posterior branch showed entrapment between the aorta and the third lumbar vertebral body (Figure 1). These findings were consistent with combined nutcracker syndrome. There was also pathological serpiginous dilatation of the left ovarian vein due to the compression of the described anterior and posterior branches of the circumaortic left renal vein with varicose pelvic veins (Figure 2).

In the venous phase, an axial CT scan demonstrated the “beak sign” of LRV, a beak angle of 40°, and a left renal vein diameter ratio of 5.7; the sagittal reconstruction demonstrated an aorto-mesenteric angle of about 21.5°.

The distance between the abdominal aorta and the third lumbar vertebral body, measured by MDCT, was 3.6 mm.

Axial CT imaging of the pelvis showed compression of the left common iliac vein by the overlying right common iliac artery in the venous phase consistent with May–Thurner syndrome without signs of venous thrombosis (Figure 3).

## 3. Literature Review

### 3.1. Materials and Methods

Through the PubMed, MEDLINE, Embase and Google Scholar database, an extensive search was performed in the fields of simultaneous combination of anterior and posterior nutcracker syndrome in the left circumaortic renal vein.

The search terms were: (“renal nutcracker syndrome”) AND (“anterior” AND “posterior”).

Only papers in English were included in the search.

### 3.2. Results

A review of the titles and abstracts was carried out. The reference lists of all identified articles were reviewed to find missed papers. This search identified 20 articles, which were then screened after reading their abstracts.

The following inclusion criteria were used: original research articles, reviews and mini-reviews, and case reports/series. These publications were carefully evaluated, considering the review’s main aims.

This evaluation left four scientific papers comprising only case reports (Table 1).

The study population, including our case, was composed of five patients, two males and two females, with a mean age of 31.4 years (d.s. ± 10.2). Flank pain was the most prevalent symptom (80%).

The type of treatment was reported in only one case (gonadal vein embolization and stenting of the major posterior renal vein).

## 4. Discussion

Variants of the renal venous system are caused by abnormal embryological development [6,10].

During fetal life, the kidneys are drained by paired dorsal and ventral veins that form a collar around the abdominal aorta, but the dorsal renal vein usually disappears during fetal development [6]. The two most common variants are the circumaortic venous ring and retroaortic left renal vein; the incidence of circumaortic LRV varies from 2 to 17% in post-mortem studies. It can also be seen as an incidental finding on 1–3% of abdominal CT scans [10,11,12].

The circumaortic venous ring is characterized by two LRVs passing anteriorly and posteriorly to the aorta, respectively [5]. In this setting, the simultaneous compression of the anterior LRV and the posterior LRV is referred to as “combined nutcracker syndrome” [13], which is a rare condition found in only four previously published reports.

More in detail, De Visschere et al. first described this condition in a 40-year-old female patient [6], and Özkan et al. described the first case of combined nutcracker syndrome in an adolescent age—a 16-year-old patient with flank pain [7].

In addition, a left crossed fused renal ectopia (CFRE), with an L-shaped kidney type, can produce a double nutcracker syndrome, as reported in Pupca et al. [8]. A case of endovascular approach in a 36-year-old woman with flank, pelvic pain, and hematuria was reported; this patient, showing an anterior and three posterior renal veins, was treated with gonadal vein embolization and stenting of the major posterior renal vein [9].

Nutcracker syndrome, first described in 1937 by Grant [14], is a condition that affects young to middle-aged patients and is more common in female patients, without an exact incidence defined [2].

Symptoms result from increased pressure in the LRV and include microhematuria, orthostatic proteinuria, and left-side flank pain secondary to the passage of ureteral blood clots [2]. Massive gonadal vein reflux may result in left-sided varicoceles in males and left-sided vulvar and pelvic varicose veins in females with symptoms of pelvic congestion syndrome (PCS) [2,15].

MDCT or MRI performed in the venous phase can be used to study the anatomy of the LRV and its relationships with the surrounding structures [1].

Coronal and sagittal Multi-Planar Reformatting (MPR) images also allow the characterization of the left gonadal vein and collateral circulation with lumbar veins.

MDCT allows the assessment of morphologic criteria of nutcracker syndrome (NS): with sagittal reconstruction, it is possible to evaluate the aorto-mesenteric angle (AMA), which is the angle between the aorta and the SMA, ranges typically ranging from 38° to 65° [1,2], an AMA lower than 35° is compatible with the diagnosis of NS; on the axial images, it is possible to recognize the beak sign, which is the stenosis of the LRV between the aorta and the SMA, to measure the beak angle, which is significant for the diagnosis of NS if it is <32°, and the LRV diameter ratio (hilar to aorto-mesenteric ratio); a value more than 4.9 has a positive predictive value of 100% for NS [16]. There are no defined angle criteria for the posterior NS [1].

CT can also depict the resulting gonadal vein dilatation and pelvic varicose veins, although it does not allow measuring flow velocity and direction [13].

Duplex Ultrasound (US) can provide practical information in these patients with a significantly higher ratio of peak systolic velocity between the point of compression and the hilar LRV [1].

A peak systolic velocity ratio of over 4.7 has been reported to have a sensitivity of 100% and a specificity of 90% for the diagnosis [17].

The definitive diagnostic test for nutcracker syndrome remains retrograde venography, despite being more invasive, which depicts the level of compression and allows the determination of the removal pressure gradient, which is diagnostic for NCS if over 3 mmHg [13].

In NS patients, management is variable, conservative or invasive, depending on the symptom severity and clinical signs [18].

Surgical management includes medial nephropexy, renal vein bypass, transposition of the left renal vein, transposition of the superior mesenteric artery, gonadal vein to inferior vena cava bypass, and auto-transplantation of the left kidney [19].

The endovascular approach with percutaneous LRV stenting has now assumed a prominent role in patients with pelvic congestion syndrome [15,20].

May–Thurner syndrome (MTS), described by the Austrian surgeon and pathologist in 1957 [21], consists of obstruction of the left CIV between the overlying right common iliac artery anteriorly and the fifth lumbar vertebra posteriorly [2].

The true prevalence of MTS is unknown, and its affects young and middle-aged people and is more common in women [2]; it may be asymptomatic or lead to swelling of the left lower extremity caused by either underlying iliofemoral thrombosis or chronic venous congestion [2]. Other symptoms include varicosities, chronic venous stasis ulcers, pulmonary embolism, or phlegmasia cerulea dolens [2].

Contrast-enhanced CT, on venous and delayed phases, is the best imaging modality in the case of suspected MTS [22], allowing for visualization of the iliac vein compression, the extent of iliofemoral vein thrombosis (if present) and excluding other causes of venous compression such as pelvic masses or lymphadenopathy [1,2].

In the absence of deep venous thrombosis, conservative management is indicated. In the case of symptoms, endovascular stent placement is considered the treatment of choice [2].

There are three previously published reports in the literature about the association between single nutcracker syndrome and May–Thurner syndrome. In one of these cases, a 26-year-old female with left lumbar pain and dysmenorrhea initially underwent variceal gonadal vein embolization and, after three months, LRV stenting [23].

We report of concomitant “combined nutcracker syndrome” and “May–Thurner syndrome” not previously described in the literature.

## 5. Conclusions

Vascular compressions by other anatomical structures are not rare, but they are often completely asymptomatic and found incidentally during tests performed for different reasons; specific compressions are described as “syndromes”.

It very rarely happens that in the same subject, there are concurrent “simple” vascular compression syndromes (association between nutcracker and others).

## Figures and Tables

**Figure 1 diagnostics-13-01433-f001:**
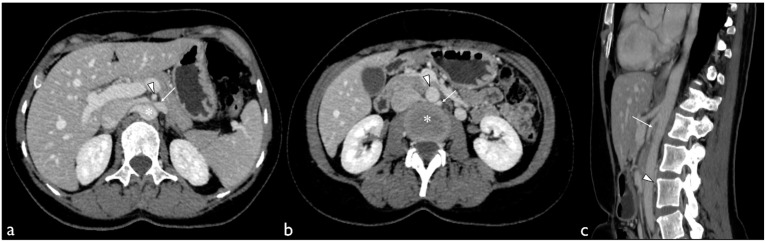
(**a**) Axial CT image showing a marked narrowing of the anterior superior branch of the LRV (arrow) as it courses between the superior mesenteric artery (arrowhead) and the abdominal aorta (asterisk). (**b**) Axial CT image showing the posterior inferior branch of the LRV (arrow) compressed between the abdominal aorta (arrowhead) and the third lumbar vertebral body. (**c**) Maximum intensity projection (MIP) image sagittal reconstruction showing the anterior superior branch (arrow) compressed between the superior mesenteric artery and the aorta and the posterior inferior branch (arrowhead) entrapped between the aorta and the vertebral body.

**Figure 2 diagnostics-13-01433-f002:**
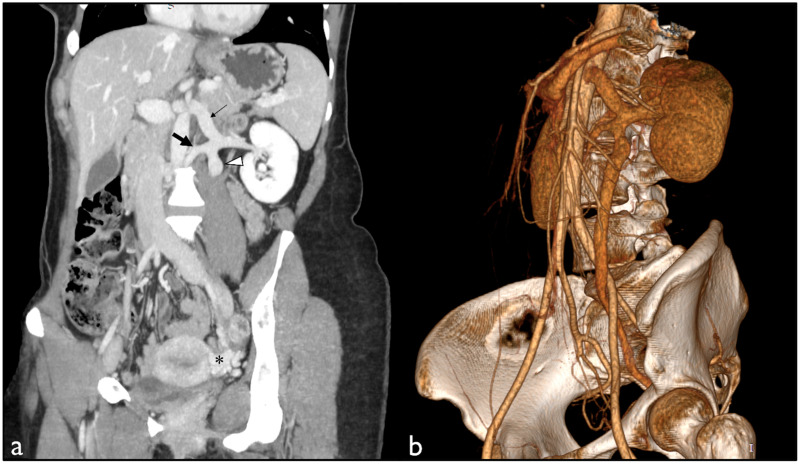
(**a**) Maximum intensity projection (MIP) image coronal reconstruction (**a**) and three-dimensional VR image (**b**) showing the circumaortic LRV, with an anterior superior branch (thin arrow) and a posterior inferior branch (thick arrow), the left gonadal vein dilatation (arrowhead) with pelvic varices (asterisk).

**Figure 3 diagnostics-13-01433-f003:**
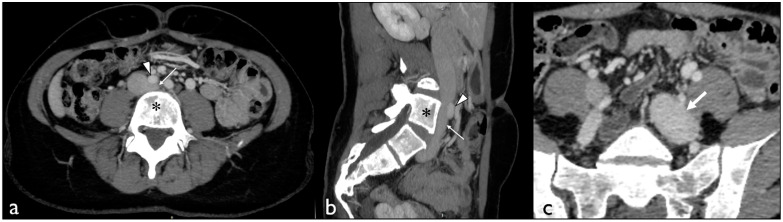
Axial CT image of the pelvis (**a**) and MIP image sagittal reconstruction (**b**) showing compression of the left CIV (arrow) between the right CIA (arrowhead) and the underlying vertebral body (asterisk). Note the upstream dilation of the common left iliac vein without thrombosis (**c**).

**Table 1 diagnostics-13-01433-t001:** Summary of the previously published cases of combined anterior and posterior nutcracker syndrome.

First Author, Year Published	Sex	Age	Anatomical Variants	Symptoms	Type of Treatment
De Visschere et al., 2008 [6]	Female	40	Circumaortic LRV	Incidental finding	-
Özkan et al., 2016 [7]	Male	16	Circumaortic LRV	Flank pain and tenderness	-
Pupca et al., 2014 [8]	Male	26	Circumaortic LRV in Crossed fused renal ectopia	Intermittent flank pain	-
Belczak et al., 2022 [9]	Female	36	One anterior and three posterior renal veins	Flank and pelvic pain and hematuria	Gonadal vein embolization and stenting of the major posterior renal vein
Tiralongo et al., 2023	Female	39	Circumaortic LRV	Intermittent flank pain	-

LRV = left renal vein.

## Data Availability

The data presented in this study are available on request from the corresponding author.

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
