# Peer review of "Anterior and Posterior Nutcracker Syndrome Combined with May–Thurner Syndrome: First Report of This Unique Case"

_diagnostics, 2023, doi:10.3390/diagnostics13081433_

Round 1

Reviewer 1 Report

line 21 her instead of their

line 25 dilation instead of dilatation

line 25 and varicose pelvic veins 

line 25 Axial CT imaging

line 26 showed instead of shows

line 28 a combination instead of the combination

line 50 "The presence of a circumaortic venous ring with a simultaneous compression..."

line 50 between the abdominal aorta and the SMA

line 51 in the literature

line 60 complained of pain in her mid-back....

line 64 hepatic instead of liver

line 65 hepatic MRI

line 65 thoracic and lumbar spine

line 66 skeletal scintigraphy

line 67 inferior vena cava

line 72 varicose pelvic veins

line 73 demonstrated

line 74 demonstrated

line 76 axial CT imaging showed compression.... in the venous phase.

line 79 axial CT image showing a marked narrowing...

line 80 axial CT image showing...

line 82 MIP showing...

line 86 showing

line 91 showing

line 92 dilation

line 100-101 "varies from 2 to 17% in post-mortem studies. It can also be seen as an incidental finding on 1-3% of abdominal CT scans"

line 103 "the simultaneous compression of the.... is referred to as combined nutcracker syndrome"

line 105 We searched the MEDLINE, Embase and Google Scholar databases for literature....

line 106 ..."and found only four previously published reports on this condition"

line 108 "first case of combined nutcracker syndrome in adolescent age - a 16-year-old patient with flank pain"

line 112 Table 1. Summary of the previously published cases of combined anterior and posterior nutcracker syndrome

line 118 left-sided flank pain

line 122 anatomy of the LRV and its relationship....

line 126 "MDCT allows the assessment..."

line 127 typically ranging from...

line 131 there are no defined angle criteria for posterior NS

line 141 for NCS if over 3mmHg

line 142 depending on the symptom severity

line 156 allowing for visualization

line 160 There are three previously published reports in the literature...

line 163 We report a case of concomitant.... not previously described in the literature. 

line 168 incidentally intead of casually

line 171-172 delete (too colloquial and informal)

line 180 this is not a retrospective analysis, but a case report. Was consent obtained from the patient (if alive) or the patient's next of kin to collect, describe and publish her medical data?

Author Response

We have modified the manuscript as you suggested

Reviewer 2 Report

Your case describes a unique pathology and therefore it would be necessary to include more information regarding your patient's past and present medical history: heredo-collateral antecedents, other associated pathologies, present and past lab tests, chronic therapy, if any. Furthermore, you should include some data related to the treatment management, once you established the diagnosis, and the prognosis of your patient. In addition, the paragraph between lines 105-111 (including Table 1) describes in fact the methodology of your work and should be deleted from the Discussion - please insert the Methodology section according to the Journal instructions. For a better understanding of this case, more data regarding nutcracker syndrome should be included: history, epidemiology, diagnosis and treatment possibilities and overall outcome. Please also explain the acronyms used in Table 1 (below the table as a note).

Author Response

  1. “Your case describes a unique pathology and therefore it would be necessary to include more information regarding your patient's past and present medical history: heredo-collateral antecedents, other associated pathologies, present and past lab tests, chronic therapy, if any”.

We include other information regarding your patient's past and present medical history; 

  1. “Furthermore, you should include some data related to the treatment management, once you established the diagnosis, and the prognosis of your patient”.

Due to the patient’s oncological disease, she didn’t start treatment management.

  1. “In addition, the paragraph between lines 105-111 (including Table 1) describes in fact the methodology of your work and should be deleted from the Discussion - please insert the Methodology section according to the Journal instructions”.

We described the methodology of our work, including a new paragraph “Literature Review”, that included materials and method, and results

  1. “For a better understanding of this case, more data regarding nutcracker syndrome should be included: history, epidemiology, diagnosis and treatment possibilities and overall outcome”. 

We included more data regarding nutcraker syndrome.

  1. “Please also explain the acronyms used in Table 1 (below the table as a note)”.

We explained the acronyms LRV (= left renal vein) below the table as a note

Round 2

Reviewer 2 Report

You performed all the recommended modifications and consequently your article has been improved.